# Genome-wide meta-analysis identifies eight new susceptibility loci for cutaneous squamous cell carcinoma

Kavita Y. Sarin [1]*, Yuan Lin [2,13], Roxana Daneshjou[1,13], Andrey Ziyatdinov[3], Gudmar Thorleifsson[4], Adam Rubin[1], Luba M. Pardo[5], Wenting Wu[2], Paul A. Khavari [1], Andre Uitterlinden [6], Tamar Nijsten [5], Amanda E. Toland [7], Jon H. Olafsson[8,9], Bardur Sigurgeirsson[8,9], Kristin Thorisdottir[8,9], Eric Jorgensen [10], Alice S. Whittemore[11], Peter Kraft [3], Simon N. Stacey [4], Kari Stefansson [4,9], Maryam M. Asgari [12] & Jiali Han[2]*

Cutaneous squamous cell carcinoma (SCC) is one of the most common cancers in the United States. Previous genome-wide association studies (GWAS) have identified 14 single nucleotide polymorphisms (SNPs) associated with cutaneous SCC. Here, we report the largest cutaneous SCC meta-analysis to date, representing six international cohorts and totaling 19,149 SCC cases and 680,049 controls. We discover eight novel loci associated with SCC, confirm all previously associated loci, and perform fine mapping of causal variants. The novel SNPs occur within skin-specific regulatory elements and implicate loci involved in cancer development, immune regulation, and keratinocyte differentiation in SCC susceptibility.

[1] Department of Dermatology, Stanford University School of Medicine, 450 Broadway St, C-229, Redwood City, CA 94305, USA. [2] Department of Epidemiology, Richard M. Fairbanks School of Public Health, Melvin & Bren Simon Cancer Center, Indiana University, 1050 Wishard Blvd, Indianapolis, IN 46202, USA. [3] Program in Genetic Epidemiology and Statistical Genetics, Harvard T.H. Chan School of Public Health, Boston, MA 02115, USA. [4] deCODE genetics/Amgen Inc., Sturlugata 8, 101 Reykjavik, Iceland. [5] Department of Dermatology, Erasmus University Medical Center, Dr. Molewaterplein 40, 3015 GD, Rotterdam, The Netherlands. [6] Department of Internal Medicine, Erasmus University Medical Center, Dr. Molewaterplein 40, 3015 GD, Rotterdam, The Netherlands. [7] Departments of Cancer Biology and Genetics and Department of Internal Medicine, Division of Human Genetics, Comprehensive Cancer Center, Ohio State University, 460W. 12th Ave, Columbus, OH 43420, USA. [8] Landspitali-University Hospital, Skaftahild 24, 105 Reykjavik, Iceland. [9] Faculty of Medicine, University of Iceland, Vatnsmyrarvegur 16, 101 Reykjavik, Iceland. [10] Division of Research, Kaiser Permanente Northern California, Oakland, CA, USA. [11] Departments of Epidemiology and Population Health and of Biomedical Data Sciences, Stanford University School of Medicine Redwood Bldg, T204, Stanford, 94305 CA, USA. [12] Department of Dermatology, Massachusetts General Hospital, 50 Staniford Street, Suite 270, 02114 Boston, MA, USA. [13]These authors contributed equally: Yuan Lin, Roxana Daneshjou. *email: ksarin@stanford.edu; jialhan@iu.edu

utaneous squamous cell carcinoma (SCC) is one of the most common cancers with an estimated 700,000 cases diagnosed in the USA annually. Metastatic SCC is responsible for 3900–8800 deaths annually in the USA[1,2]. Risk factors for SCC include age, gender, fair skin pigmentation phenotypes, ultraviolet radiation exposure, and immunosuppression[3]. While the risk factors for SCC development have largely been attributed to environmental exposures and skin pigmentation, there has been a growing appreciation of the contribution of germline genetics in SCC development.

Recently, three genome-wide association studies (GWAS) have identified 14 single-nucleotide polymorphisms (SNPs) associated with cutaneous SCC[4–6]. These studies include a GWAS in 7404 SCC cases and 292,106 controls in the 23andMe, the Nurses' Health Study (NHS) and the Health Professionals Follow-Up Study (HPFS) cohort[4], a GWAS in 7701 SCC cases and 60,186 controls from the Kaiser Permanente Northern California healthcare system[6], and a GWAS in 745 SCC cases and 12,805 controls from Rotterdam Study, NHS, and HPFS[5]. These 14 SNPs involve loci which affect skin pigmentation, but also occur in loci associated with cell-mediated immunity, anti-apoptotic pathways and cellular proliferation.

Unfortunately, further identification of SCC risk loci has been hampered by a lack of well-phenotyped cohorts and a cancer registry for cutaneous SCC. To aid in this, we developed a SCC-GWAS consortium comprised six international cohorts with data on cutaneous SCC. Here, we present the results of the largest cutaneous SCC meta-analysis to date, totaling 19,149 SCC cases and 680,049 controls. We discover eight novel loci associated with cutaneous SCC, confirm all previously associated loci, and perform fine mapping of causal variants. The novel SNPs occur within skin-specific regulatory elements and implicate loci involved in cancer development, immune regulation, and keratinocyte differentiation in SCC susceptibility.

## Results and discussion
**Cohort description**. The GWAS meta-analysis consisted of 19,149 SCC cases and 680,049 controls, including 2081 SCC cases and 296,015 controls from deCODE genetics in Iceland, 398 cases and 10,629 controls from Rotterdam, Netherlands, 6579 cases and 280,558 controls from 23andMe, 2287 cases and 30,966 controls from NHS/HPFS, 103 cases and 1715 controls from Ohio State University Hospital, and 7701 cases and 60,166 controls from Kaiser Permanente. Demographics and further details on these studies are found in the "Methods" and Supplementary Tables 1 and 2.

**Genome-wide significant novel susceptibility loci**. This meta-analysis reinforced all 14 previously described loci associated with cutaneous SCC (Fig. 1; Supplementary Table 3). Recently a C-terminal exon mutation in the *BRCA2* gene (K3326*, rs11571833) was reported to confer risk of SCC[7]. We examined the meta-analysis data and found that rs11571833 is associated with SCC with an effect size of 0.36 (log odds ratio) for the alternate (minor) allele and *p*-value $1.0 \times 10^{-6}$, confirming the reported observation and highlighting the contribution of DNA repair genes to SCC risk.

In addition to confirming all previous susceptibility loci (Fig. 1; Supplementary Table 3), this meta-analysis identified eight novel susceptibility loci for cutaneous SCC: rs10399947 (1q21.3), rs10200279 (2q33.1), rs10944479 (6q15), rs7834300 (8q23.3), rs1325118 (9p23), rs7939541 (11p15.4), rs657187 and rs11170164 (12q13.13), rs721199 (12q23.1) (Table 1; Supplementary Tables 4, 5). Forest plots of the individual GWAS study results are detailed in Supplementary Figs. 3A–3V. Regional association plots are

found in Supplementary Figs. 4A–4V. These loci included genes involved in cancer progression (*SETDB1*: rs10399947, *CASP8/ALS2CR12*: rs10200279, *WEE1*: rs7939541), immune regulation (*BACH2*: rs10944479), keratinocyte differentiation (*TRPS1*: rs7834300, *KRT5*: rs11170164 and rs657187), and pigmentation (*TYRP1*: rs1325118). These loci are discussed in detail below.

**Fine-mapping resolution at the associated loci**. We sought to refine the localization of potential functional variants in the 22 genome-wide significant loci using a Bayesian approach (Methods). Conditional analyses in 18 of the 22 identified loci revealed 21 distinct association signals or index SNPs with $p < 5 \times 10^{-8}$ (Supplementary Table 6, Supplementary Table 7). We further estimated 99% the credible sets for every index SNP in 18 loci. We excluded two loci from conditional analysis: the locus 6p21.32 was excluded as this is an HLA locus. The *MC1R* locus at 16q24.3 showed evidence of a large number of SNPs (24) driving the association, suggesting, in part the presence of allelic heterogeneity[8]. This is consistent with previous studies including a recent GWAS in the UK Biobank, which found 31 SNPs independently associated with red hair color near *MC1R*, of which only 10 were coding variants[9,10]. Due to allelic complexity and potential artifacts with an external LD reference panel, this locus was also excluded from conditional analysis. We found that the number of SNPs in the sets across 18 loci ranges from 1 to 1990 with a mean value of 136. The lead SNP at seven signals accounted for >0.80 of posterior probability of association (PPA, Methods) and, at six of these signals including rs7939541 in the novel 11p15.4 locus, PPA exceeded 0.99.

Fine mapping revealed three loci with distinct secondary signals: rs6935510, rs10962599, and rs4778138. rs6935510 at locus 6p25.3 ($r^2 = 0.12$ from the lead SNP rs12203592 in CEU population) is 2 kb upstream of IRF4 in a predicted bivalent promoter region and alters a number of regulatory motifs. IRF4 is a transcription factor downstream of MITF and is associated with photosensitivity, freckles, blue eyes, and brown hair color[11]. rs10962599, an intronic variant in the skin pigmentation gene BNC2 at 9p22.2, independent from lead SNP rs10810657 ($r^2 = 0.0012$ in CEU population) and in a H3K4me1 enhancer region in melanocytes. rs4778138 at 15q13.1 is independent from the lead SNP rs1800407 ($r^2 = 0.0032$ in CEU population). rs4778138 is an intronic variant in a novel locus, OCA2, and has been implicated in melanoma risk, hair and eye color[12–14].

**SNPs associated with pigmentation and photodistributed sites**. Fair skin and sun exposure are well-described risk factors for SCC. We analyzed the 22 SCC risk loci for an association with pigmentation phenotypes in the deCODE cohort, including eye color, hair color, freckling, and photosensitivity (Supplementary Table 8). Pigmentation information was self-reported as previously described[15,16]. Nine out of 22 index SNPs were associated with pigmentation phenotypes, including two novel SNPs; rs7834300, an intronic SNP in *TRPS1* associated with sun sensitivity, and rs1325118, located 66 kb upstream of *TYRP1* and is associated with eye color[17].

Although sun exposure information was not available for the majority of cohorts, we sought to determine potential gene–environment interactions by performing a site-stratified analysis of SCC risk loci to determine SNPs associated with SCC in photodistributed sites. Cohorts with SCC site information (deCODE, NHS/HPFS, Rotterdam, and Ohio) were divided into high photoexposure (head and neck, upper extremities) and low photoexposure sites (trunk and legs) based on site location of the first SCC. We observed one SNP, rs721199, in which the T allele was specifically protective against SCC in low-photodistributed

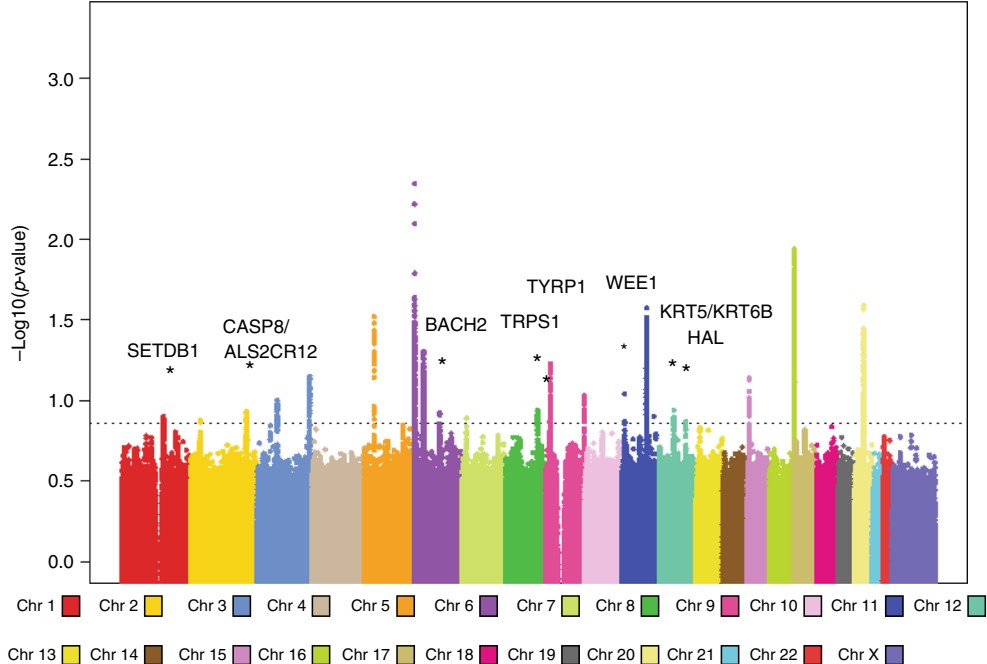

**Fig. 1 Manhattan plot of the combined meta-analysis of GWAS of SCC.** The $P_{fixed}$ Stage one value for all SNPs present in at least two studies have been plotted using a $-\log_{10}(p\text{-value})$. The total Stage one meta-analysis included eight SCC GWAS, totaling 19,149 cases and 680,049 controls. $p < 5 \times 10^{-8}$ (genome-wide significance) threshold is indicated by a dashed line. In total, 22 loci reached genome-wide significance, including 8 novel loci 1q21.3, 2q33.1, 6q15, 8q23.3, 9p23, 11p15.4, 12q13.3, and 12q23.1 are highlighted by *.

**Table 1 Novel associations in SCC-GWAS meta-analysis.**

| SNP | Chr | Position | Locus | Gene | Major allele | Minor allele | MAF | Odds ratio (95% CI) | Direction | *p*-value |
|-----|-----|----------|-------|------|--------------|--------------|-----|---------------------|-----------|-----------|
| rs10399947 | 1 | 150861960 | 1q21.3 | ARNT--[]--SETDB1 | G | A | 0.368 | 0.94 (0.92–0.96) | −, −, −, −, −, + | 6.65E-09 |
| rs10200279 | 2 | 202170655 | 2q33.1 | [ALS2CR12] | C | T | 0.287 | 1.07 (1.05–1.10) | +,+,+,+,+, + | 2.67E-09 |
| rs10944479 | 6 | 90880393 | 6q15 | [BACH2] | G | A | 0.189 | 0.91 (0.89–0.94) | −, −, −, −, −, −, N | 3.75E-09 |
| rs7834300 | 8 | 116611632 | 8q23.3 | [TRPS1] | C | G | 0.438 | 1.07 (1.05–1.09) | +,+,+,+, −,+ | 2.01E-09 |
| rs1325118 | 9 | 12619616 | 9p23 | []--TYRP1 | T | C | 0.304 | 0.94 (0.91–0.96) | −, −, −, −, +, − | 4.38E-08 |
| rs7939541 | 11 | 9590389 | 11p15.4 | ZNF143--[]--WEE1 | T | C | 0.410 | 1.08 (1.06–1.10) | +,+,+,+,+, + | 9.23E-12 |
| rs657187 | 12 | 52898985 | 12q13.13 | KRT6A--[]--KRT5 | A | G | 0.420 | 0.93 (0.92–0.96) | −, −, −, −, −, − | 1.80E-09 |
| rs721199 | 12 | 96374057 | 12q23.1 | [HAL] | C | T | 0.463 | 0.94 (0.92–0.96) | −, −, −, −, −, − | 3.55E-08 |

MAF: minor allele frequency, CI: confidence interval, build GRCh37. [] represents location of SNP either in relationship to known genes with [gene] indicating SNP is within the gene and gene—[]— indicating intergenic SNPs. Minor allele is effect allele. Minor allele frequency (MAF) is based on the pooled meta-analysis. Direction is listed in order for 23me, deCODE, NHS/HPFS, Kaiser, Ohio, and Rotterdam. N means not included in analysis.

sites (Supplementary Table 9). rs721199 is an eQTL in skin tissue for *HAL* (sun-exposed lower leg skin, $p = 4.1 \times 10^{-79}$ and sun-exposed suprapubic skin $1.2 \times 10^{-67}$) which has been shown to play a role in UV radiation mediated immunosuppression. This highlights a potential gene–environment interaction which contributes to SCC development.

**Heritability of SCC.** We estimated the overall contribution of common variants to SCC risk using LD Score Regression[18]. Approximately 25% (95% confidence interval 0.17–0.32) of the familial relative risk for SCC can be explained by common variants across the genome. In contrast, the 22 genome-wide significant loci explain 8.5% of the familial relative risk. This suggests that there are additional SCC risk loci that could be identified in a larger GWAS. We also used LD Score Regression to explore whether particular regions of the genome disproportionately contributed to the overall common-variant heritability. We partitioned common-variant heritability across 53 publicly available, non-cell-type-specific annotations and observed significant enrichment in heritability (FDR < 0.1) for

coding regions (6.7 × enrichment, $p = 8.5 \times 10^{-4}$), super enhancers (2.1×, $p = 1.2 \times 10^{-3}$), and H3K4me3 histone promoter marks (1.7 × $p = 5.5 \times 10^{-3}$). Heritability in repressed regions was significantly depleted (0.5×, $p = 8.5 \times 10^{-3}$) (Supplementary Table 10)[19]. We also conducted enrichment analyses using 220 cell-type-specific histone marks; none of these marks were significantly enriched (Supplementary Table 11)[19]. These findings highlight the increased contribution to SCC risk from variants, which affect protein coding and gene regulation.

**Description of novel loci.** At 1q21.3, rs10399947 has a PPA of 0.02, and is an eQTL for multiple genes in skin tissue, including *SETDB1*, *ECM1*, and *CERS2* (Supplementary Table 12). *SETDB1* encodes a histone methyltransferase and is associated with the propagation of several malignancies, including melanoma[20,21]. *ECM1* codes for the extracellular matrix protein 1, and has been found to be overexpressed in epithelial malignancies as well as melanoma cell lines[22,23]. *CERS2* encodes ceramide synthase 2 and is thought to inhibit metastases and invasion across multiple cancer types, including breast cancer[24].

At 2q33.1, rs10200279 has a PPA of 0.12 and is an intronic SNP of *ALS2CR12*, an eQTL in skin tissue for *CASP8*, *ALS2CR12*, *CASP10*, and *PPIL3* and alters six regulatory motifs (Supplementary Table 12)[25,26]. The *CASP8/ALS2CR12* locus has been implicated in multiple cancer types, including basal cell carcinoma and breast cancer[27–29]. *CASP10* is a homologue for *CASP8* and has been found to inhibit tumorigenesis; loss-of-function mutations have been reported in multiple cancer types. *PPIL3* is proximal to *CASP8* and has been independently associated with estrogen receptor-negative breast cancer[30]. rs10200279 is LD with rs700635 (PPA 0.08, $r^2 = 0.97$ in European 1000G Phase 1 population), which has been associated with basal cell carcinoma risk and shown to functionally affect splicing of the cellular apoptosis regulator, *CASP8*[27,29,31]. Ten SNPs had a PPA threshold of 0.05 and could also represent potential causal variants. These are listed in Supplementary Table 13. Interestingly, all of them are eQTLs in the skin tissue for CASP8 and ALS2CR12.

At 6q15, rs10944479 has a PPA of 0.29 and is an intronic SNP of *BACH2*, which encodes a transcription factor involved in tumor immunosuppression and response to anti-PD-1 treatment[32,33]. This SNP alters two predicted regulatory motifs (HNF6 and Hoxa10)[17]. Expression of *BACH2* was suppressed by 57% in SCC as compared with paired matched normal skin ($p = 6.8 \times 10^{-9}$) highlighting a potential mechanism by which SCC could evade immune surveillance (Fig. 2).

At 8q23.3, rs7834300 has a PPA of 0.05 and is an intronic variant in *TRPS1*, a sequence-specific transcriptional repressor important for bone, hair follicle, and kidney differentiation. Recently, *TRPS1* has been associated with tanning response[34]. rs7834300 alters two regulatory motifs (GR, Zec)[17,25]. In the deCODE cohort, this variant was associated with sun sensitivity (Supplementary Table 8).

At 9p23, rs1325118 has a PPA of 0.5 in our analysis and is 66 kb upstream of *TYRP1*, a pigmentation gene and alters three predicted regulatory motifs. In the deCODE cohort, rs1325118 was also associated with eye pigmentation (Supplementary Table 8)[35]. In

SCC samples, expression of *TYRP1* was suppressed 58% as compared with matched normal skin biopsies ($p = 3 \times 10^{-5}$), suggesting that keratinocytes in SCC may have defects in differentiation and contain reduced pigmentation (Fig. 2).

At 11p15.4, rs7939541 accounts for over 99% of the PPA at this locus and is 5.8 kb upstream of *WEE1*. It is in an enhancer feature and is an eQTL in skin tissue for *WEE1*, *snoU13* (Supplementary Table 12), alters two predicted regulatory motifs and is in a DNAse hypersensitivity site for multiple tissues, including the skin. This SNP falls in a region marked by H3K27ac and H3K4me1 enhancer-associated histone marks, with lack of the repressive H3K27me3 mark in primary keratinocytes (Fig. 3). In addition, WEE1 transcript levels were suppressed in SCC as to the normal skin (Fig. 3, $p = 0.0002$) *WEE1* encodes a kinase that is a G2-M checkpoint inhibitor and is highly expressed in multiple cancer types, including melanoma and non-cutaneous squamous cell carcinoma[36,37]. *WEE1* inhibition can increase the sensitivity of several different cancer types to radiation or chemotherapy[36].

At 12q13.13, rs657187 has a PPA of 0.22, is 9.4 kb 3′ of *KRT5*, and alters two predicted regulatory motifs[17,25]. It is also an enhancer feature in the skin and an eQTL of *KRT6C* in the skin (Supplementary Table 12), a keratinocyte development gene[17,38]. Expression of *KRT6C* was 8.5 times higher in SCC as compared with the normal skin (Fig. 2, $p = 5.51 \times 10^{-13}$). rs657187 is in low LD ($r^2 = 0.052$ in CEU) with rs11170164 (PPA = 0.01), a nearby SNP which encodes a G138E substitution in KRT5 and has been previously associated with BCC and SCC[39]. Conditional analysis of rs657187 and rs11170164 indicated that these variants each have independent effects ($p_{adj} = 2.28 \times 10^{-6}$ and $5.67 \times 10^{-5}$, respectively, Supplementary Table 14).

At 12q23.1, rs721199 has a PPA of 0.36, is an intronic SNP of *HAL*, alters three predicted regulatory motifs, and is an eQTL in the skin tissue for *HAL* and *RP11-256L6.3* (Supplementary Table 12)[17,25]. *HAL* is highly expressed in the skin and plays a role in UV-mediated immunosuppression[40]. In the stratification analysis by photodistributed site (high or low), the protective

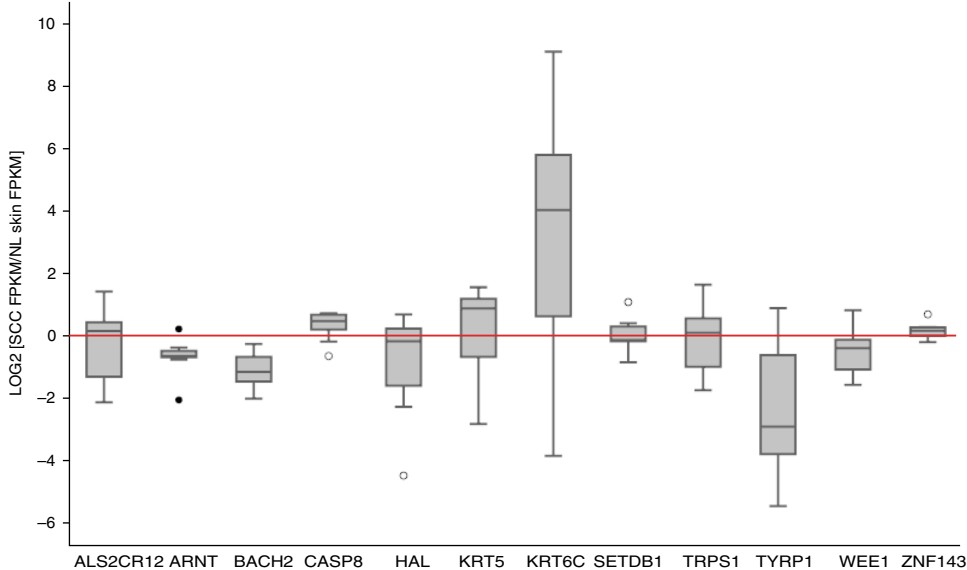

**Fig. 2 Gene expression analysis for novel SCC susceptibility loci.** RNA-seq data were obtained from Gene Expression Omnibus (GSE84194) were analyzed by DESeq. Transcript levels (FPKM) in SCC samples were compared with levels in paired matched normal skin. Boxplot demonstrates log₂[SCC/ Normal skin] expression levels for 13 genes surrounding the novel SNPs. Legend for box and whisker plots. The black center line denotes the median value (50th percentile), while the gray box contains the 25th to 75th percentiles of data set. The black whiskers mark the 5th and 95th percentiles, and values beyond these upper and lower bounds are considered outliers, marked with white circles. The red threshold line indicates the point where these is no change in gene expression between SCC tumor and normal skin. *ARNT, BACH2, TYRP1,* and *WEE1* were significantly downregulated in SCC as compared with normal skin and *CASP8 and KRT6C* were upregulated in SCC relative to normal skin by DESeq.

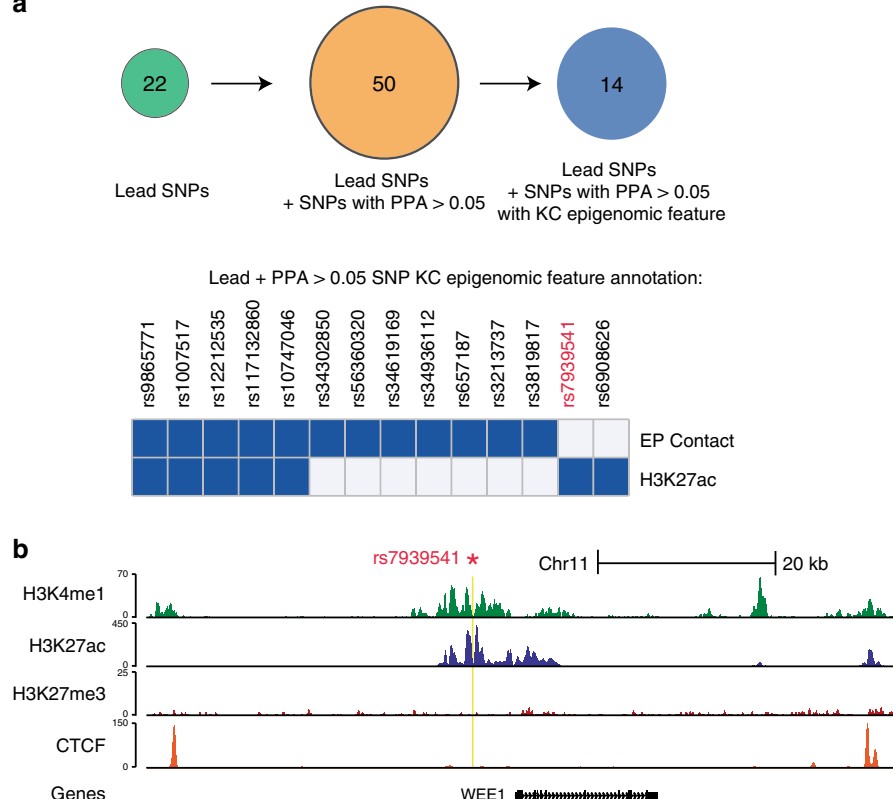

**Fig. 3 Annotation of novel SNPs with epidermal enhancer information. a** Top: Circles represent the number of SNPs considered at each stage of the workflow to identify epigenetic context of all novel SNPs. We started with 22 lead SNPs identified by meta-GWAS, then found putative causal SNPs defined as any SNPs with a PPA of >0.05 from our fine-mapping analysis. We next refined that expanded list to SNPs for which the genomic location overlapped a previously identified epigenomic feature (either the H3K27ac enhancer mark or ends of an enhancer–promoter contact). Bottom: Heatmap displaying the overlap of SNPs with enhancer–promoter contacts or H3K27ac marked regions. The blue designation indicates that the SNP overlaps at least one H3K27ac region or contact. **b** Genome browser tracks for the genomic locus for SCC-index SNP rs793954, PPA > 0.99, demonstrating enhancer features in primary human keratinocytes (KC). ChIP-seq signal tracks are displayed for H3K4me1 and H3K27ac (which typically mark active enhancers and promoters) as well H3K27me3 (which marks inactive loci). Yellow denotes SNP location; note this SNP falls in a region marked by H3K27ac and H3K4me1 enhancer-associated histone marks, with lack of the repressive H3K27me3 mark. CTCF sites indicate that the SNP is not involved in CTCF loops and associated TADs.

association with rs721199 T allele occurred only in the low-photodistributed site, and the heterogeneity in the effect sizes among the subgroups was significant ($p = 0.03$, Supplementary Table 9). The T allele is associated with higher expression levels of *HAL*[26].

**Conclusion.** In conclusion, this GWAS meta-analysis of 19,149 cases and 680,049 controls from the USA and Europe represents a threefold increase in sample size compared with the previous SCC-GWAS studies, and reinforced all 14 previously reported loci. In addition, this meta-analysis identified eight novel susceptibility loci. In total, the 22 loci explain 8.5% of heritable risk for SCC. Subanalyses of these 22 loci identify 9 loci associated with pigmentation phenotypic traits and 1 locus (*HAL*) associated with photodistribution-specific risk. In addition, fine mapping identifies potentially causal SNPs which fall within putative regulatory elements in keratinocytes and melanocytes and regulate the expression of genes involved in cancer progression, differentiation, and immune regulation, highlighting the role of these pathways in modulating SCC susceptibility.

## Methods

**Study design.** The GWAS meta-analysis is comprised six international cohorts (Supplementary Table 1). The GWAS data set from the personal genetic company 23andMe Inc. encompassed 6579 SCC cases and 280,558 controls of European

ancestry who consented to participate in research. The GWAS data set from the Nurse's Health Study (NHS)/ Health Professionals Follow-Up Study (HPFS) consisted of 2287 SCC cases and 30,966 controls of European ancestry. The 23andMe data and some of the NHS/HPFS data were used in a previously reported GWAS[4]. Kaiser Permanente Northern California contributed a GWAS set encompassing 7701 cases with incident SCC and 60,166 controls of European ancestry. Some of these data were used in another previously reported GWAS[5,6]. GWAS data from the deCODE study encompassed 2081 SCC cases and 296,015 controls of European ancestry. The Rotterdam study contributed a GWAS data set consisting of 398 cases with SCC and 10,629 controls of European ancestry. The Ohio study included GWAS data on 103 SCC cases and 1715 controls of European ancestry. Supplementary Table 2 shows the gender and age of cases and controls from each cohort.

**Case validation.** Cases were medically adjudicated for the NHS/HPFS, Kaiser, Rotterdam, and Ohio cohorts by histopathologic records. The deCODE cases were ascertained from the Icelandic Cancer Registry, and were all histopathologically confirmed. Cases were self-reported in the 23andMe cohort. In the self-reported cases, survey response accuracy was validated by comparing a subgroup of survey responses with medical record data, which revealed a sensitivity and specificity of 92 and 98%, respectively[4].

**Genotyping.** All samples were collected with informed consent and ethical oversight. Samples were genotyped on a variety of commercial arrays, as previously detailed[4–7,15].

**Quality control and imputation.** All cohorts underwent strict quality control (QC) procedures and were imputed using the following reference panels: Kaiser cases were imputed using the 1000 Genomes Phase 1 integrated release, March 2012, with Aug 2012 chromosome X update, with singletons removed. 23andMe cases were imputed

using the March 2012 Version 3 release of 1000 Genomes Phase 1 reference haplotypes. NHS/HPFS cases were imputed using the 1000 Genomes Project ALL Phase 1 Integrated Release Version 3 (March 2012) Haplotypes with singletons removed. Ohio cases were imputed using 1000 Genomes Phase 3. The Rotterdam cases were imputed using the latest version of Genome of the Netherlands (GoNL) data as the reference. The deCODE data were processed using long-range phasing and imputation based on data from the Icelandic population[15]. Only variants which were found in either the 1000 Genomes Phase 1 Version 3 data set or the Haplotype Reference Consortium data set (version 1.1) were included in the deCODE data. Variants with large differences in frequency between Icelandic and European populations were excluded from the deCODE data. Further information on association analysis of individual studies has been reported previously[4,6].

**Individual genome-wide association analysis.** The methods used for association testing in each cohort have been described in detail[4–8]. Briefly, association analysis was performed using logistic regression, assuming an additive model for allelic effects. Sex and population stratification () were adjusted for by principal component (PC) analysis in each cohort, except deCODE. The deCODE cohort was adjusted differently because it utilizes familial imputation for individuals who have not been directly genotyped[7]. Five PCs were included to adjust for population stratification in the 23andMe and the NHS/HPFS cohorts. Ten PCs were adjusted for in the Kaiser and the Ohio cohorts. Rotterdam cohort adjusted for the four largest PCs. The linkage disequilibrium (LD) score regression was applied in the deCODE cohort to account for inflation in test statistics due to cryptic relatedness and stratification in the Icelandic population[18]. The $\chi^2$ statistics from GWAS scan were regressed against LD score and then the intercept was used as a correction factor[9].

**Meta-analysis.** SNPs with imputation quality $R^2 < 0.3$ in any data set were excluded from that individual study prior to meta-analysis. For each study, SNPs with low expected minor allele counts in cases (overall minor allele frequency times number of cases < 10) were also removed before meta-analysis. Fixed-effects meta-analysis was conducted using the METAL software. Heterogeneity of per-SNP effect size in each cohort contributing to overall meta-analyses was assessed using heterogeneity $I^2$ Cochran's Q statistic (Supplementary Tables 3, 4). The meta-analysis genome-wide inflation value (λ) was 1.06. QQ plots of the GWAS meta-analysis and individual study p-values are provided (Supplementary Tables 3, 4). SNPs were considered significant if they had a p-value less than $5 \times 10^{-8}$. Individual study p-values are listed in Supplementary Table 5. Effects are given as log odds ratio (β).

**Proportion of familial relative risk.** We estimated the proportion of familial relative risk due to identified, genome-wide significant variants using

$$\frac{\left[\sum_i \hat{\beta}_i^2 q_i (1 - q_i)\right]}{\ln(\lambda)}, \qquad (1)$$

where $\hat{\beta}_i$ and $q_i$ are the estimated log odds ratio and minor allele frequency for variant $i$ and $\lambda$ is the familial relative risk for SCC ($\lambda = 2.7$)[41,42]. To estimate the proportion of familial relative risks explained by tagged common variants across the whole genome, we used

$$\frac{\left[\hat{h}_{\mathrm{obs}}^2 / (P(1 - P))\right]}{\ln(\lambda)}, \qquad (2)$$

where $\hat{h}_{\mathrm{obs}}^2$ is the estimate of "observed scale" heritability obtained from LD Score Regression applied to the SCC meta-analysis summary statistics ($\hat{h}_{\mathrm{obs}}^2 = 9.3 \times 10^{-3}$, SE $= 1.5 \times 10^{-3}$, $p = 5.6 \times 10^{-10}$), and $P$ is the fraction of cases in the overall sample (2.8%).

**Functional annotation of GWAS meta-analyses.** We performed linkage disequilibrium (LD) score regression analyses using the summary statistics from the meta-analyses of the six GWASes[19]. We restricted analysis to all SNPs present on the HapMap version 3 data set that had a MAF > 1% and an imputation quality score $R^2 > 0.3$ across all studies. LD scores were calculated using the 1000 Genomes Project Phase 3 EUR reference panel. For stratified analyses taking genomic annotations into account, we created a "baseline model" model with 53 non-cell-type-specific overlapping annotations[19]. We also performed analyses using 220 cell-type-specific annotations for four histone markers (H3K4me1, H3K4me3, H3K9ac, and H3K27ac) across 27–81 cell types, depending on the histone marker[19]. For the cell-type-specific analyses, we augmented the baseline model by adding these annotations individually, creating 220 separate models, each with 54 annotations (53 + 1).

**Annotation of SNPs with epidermal enhancer site information.** The 22 genome-wide significant SNPs as well as SNPs with a posterior probability of association (PPA, Methods) > 0.05 in our fine-mapping analysis were annotated for enhancer features using our keratinocyte genome-wide promoter capture Hi-C (CHi-C) and H3K27ac ChIP-seq (Fig. 3a)[43]. Enhancer–promoter (EP) contacts and H3K27ac ChIP-seq peaks were derived from Rubin et al.[43]. SNP locations were filtered for

direct overlap with H3K27ac peaks or the ends of enhancer–promoter contacts. Contacts were annotated at 10 kb resolution, so SNPs overlapping either 10-kb window marking the ends of a contact were considered overlapped. The WashU Epigenome Browser was used to visualize a SNP and the tracks from the ENCODE Project for NHEK as well as contacts (FDR < 0.01, proximal to the SNP) from progenitor keratinocytes are displayed.

**Functional annotation of significant loci.** To further annotate regulatory function, PubMed and the NHGRI-EBI GWAS catalogue (version updated 4/10/2018) were queried for prior publications regarding SNP function and disease association[44]. We identified the closest related gene and evidence of regulatory function using HaploReg v4.1 (http://archive.broadinstitute.org/mammals/haploreg/haploreg.php)[17]. Gene annotations were based on the UCSC Genome Browser and GENCODE version 13.BEDTools was used to calculate the proximity of each variant to a gene by either annotation, as well as the orientation (3' or 5') relative to the nearest end of the gene, based on the strand of the gene. For each index SNP or linked SNP $r^2 \geq 0.8$ or SNP with a PPA >0.05, we extracted data on expression quantitative trait loci (eQTL) for sun-exposed (lower leg) and not sun-exposed (suprapubic area) skin tissue using GTEx portal dbGaP release V8[26].

**Gene expression analysis.** Raw RNA-seq data for nine paired matched SCC and normal skin samples biopsied from eight patients. One patient had two SCCs from different sites. (GSE84194 [https://www.ncbi.nlm.nih.gov/geo/query/acc.cgi?acc=GSE84194]) were obtained from the GEO (http://www-ncbi-nlm-nih-gov.laneproxy.stanford.edu/geo/)[45]. Actinic keratosis samples from this data set were excluded from analysis. Reads were aligned to the human genome (hg19) using Tophat (v2.1.1). Featurecounts (v1.5.2) was used to generate count data and Cufflinks (v2.2) to generated relative transcript levels in Fragments Per Kilobase of transcript per million mapped read (FPKM), and DESeq (v1.6.3) using a matched sample model was used to identify differentially expressed genes between the SCC and normal skin samples. Each gene of interest was selected by closest proximity to one of the eight novel risk variants; however, if a lead SNP was an eQTL in the skin tissue for a more distant gene, then this gene was chosen as well. Boxplot was used to visualize the expression of the SCC relative to normal skin of the genes surrounding the eight novel SNPs.

**Fine mapping.** We used GCTA-COJO to establish distinct association signals at the genome-wide significant loci with SCC susceptibility[46]. GCTA-COJO performs an approximate conditional analysis using association summary statistics from GWAS meta-analysis and the LD information estimated from a reference panel. For each locus, we defined a 2 Mb region encompassing 1 Mb from the lead SNP (using summary statistics) on both sides to ensure long-range genetic signals are not missed. Conditional independent variants that reach genome-wide significance level (the GCTA-COJO default level, $5 \times 10^{-8}$) were considered as index SNPs for distinct association signals. We applied additional filters to association summary statistics and discarded variants with (i) MAF < 0.1%; (ii) ambiguous A/T and G/C alleles; and (iii) allele coding and frequency mismatches between genotypes in summary statistics and LD reference panel (implemented in GCTA-COJO). We defined the effective sample size for each cohort and used these estimates further in the analysis:

$$N_{\mathrm{eff}} = 4N_{\mathrm{cases}}N_{\mathrm{controls}} / (N_{\mathrm{cases}} + N_{\mathrm{controls}}). \qquad (3)$$

We used imputed genotypes in the Harvard cohort (the imputation quality $R^2 > 0.3$) as a reference panel for LD r measures (the Pearson correlation). We selected the Harvard cohort as a reference panel for LD r measures (the Pearson correlation), because it was the largest cohort of our meta-analysis, in which we have access to raw genotype data. We used imputed genotypes with the imputation quality $R^2 > 0.3$[47]. The total number of individuals in the Harvard reference panel was 7403; the per-locus overlap between variants in summary statistics and reference panel was > 80% for variants with MAF > 0.01 and >50% for variants with MAF = 0.001–0.01. After applying the quality control, we had 19 of the 22 loci with lead SNPs passing the significance threshold ($p < 5 \times 10^{-8}$) and, thus, available for the analysis. The two discarded loci had their lead SNPs with MAF < 1%, which were filtered out likely due low coverage of the genotyping platforms or insufficient density of genotype imputation panels[47]. Another two loci in the MHC region (16p21.32) and MC1R (16q24.3) region were excluded to their complicated LD structures (Supplementary Table 6).

For each association signal from the conditional analyses by GCTA-COJO, we computed an approximate Bayes factor in favor of association on the basis of effect sizes and standard errors from the GWAS summary statistics within the 2 Mb region of the locus[48]. When loci showed a single-association signal, the summary statistics were taken from unconditional GWAS. When loci exhibited multiple association signals, the summary statistics were derived from the approximate conditional analysis adjusting for all other index variants in the region. The prior probabilities of the variant to be causal were assumed to be the same among all the variants in the region and equal to $1/M$, where $M$ is the number of variants in the region. For the $i$th variant the approximate Bayes factor is:

$$\mathrm{BF}_i = \sqrt{\frac{V_i}{V_i + \omega}} \exp\left(\frac{w\beta_i^2}{2V_i(V_i + w)}\right) \qquad (4)$$

where $\beta_i$ and $V_i$ denote the effect size and variance (the squared standard error) of the variant $i$ from unconditional or approximate conditional association analysis. The parameter $\omega$ denotes the prior variance in effects, which is set to 0.04 (Wakefield, 2007)[48].

Then the posterior probability that the $i$th variant is a true association signal (PPA) is:

$$\pi_i = \frac{\text{BF}_i}{\sum_{m=1}^{M} \text{BF}_m} \tag{5}$$

The 99% credible set is defined as the minimal number of variants with the cumulative PPA of 0.99. The procedure to compute the 99% credible set is accomplished in two steps: (i) order the variants in descending order of their PPA; (ii) include ordered variants until the cumulative PPA reaches 0.99[49].

**Stratified association analysis by photodistributed sites.** According to the approach by Lin et al.[50] we estimated the heterogeneity of genetic effect size between high- and low-photoexposure site, considering overlapping controls used in high- and low-photodistributed site cohorts[50].

Correlation of the genetic effects of per-SNP in high- and low-photodistributed site in each study is estimated by:

$$\text{Corr}(\hat{\beta}_1, \hat{\beta}_2) = \left( n_{120}\sqrt{\frac{n_{11}n_{21}}{n_{10}n_{20}}} + n_{121}\sqrt{\frac{n_{10}n_{20}}{n_{11}n_{21}}} \right) \Big/ \sqrt{n_1 n_2} = \frac{\sqrt{n_{11}n_{21}}}{\sqrt{n_1 n_2}} \tag{6}$$

Where $\hat{\beta}_1$ is the estimate of log odds ratio of an individual SNP in high-photodistributed site in each study, $\hat{\beta}_2$ is the estimate of log odds ratio of the SNP in low-photodistributed site in each study[50].

$n_{11}$, $n_{10}$ and $n_1$ are, respectively, the number of cases, the number of controls, and the total number of subjects in the high-photodistributed cohort and $n_{21}$, $n_{20}$ and $n_2$ are, respectively, the number of cases, the number of controls, and the total number of subjects in the low-photodistributed cohort.

Given that the controls in the high- and low-photodistributed site cohorts are totally overlapped, $n_{120} = n_{10} = n_{20}$; whereas the cases are not shared: $n_{121} = 0$.

The difference $\hat{\delta}$ between the genetic effects of the SNP in high- and low-photodistributed site in each study is estimated by:

$$\hat{\delta} = \hat{\beta}_1 - \hat{\beta}_2 \tag{7}$$

The variance of $\hat{\delta}$ in each study is estimated by:

$$\text{Var}\left(\hat{\delta}\right) = \text{Var}\left(\hat{\beta}_1\right) + \text{Var}\left(\hat{\beta}_2\right) - 2\text{Corr}\left(\hat{\beta}_1, \hat{\beta}_2\right)\sqrt{\text{Var}\left(\hat{\beta}_1\right)}\sqrt{\text{Var}\left(\hat{\beta}_2\right)} \tag{8}$$

Where $\text{Var}\left(\hat{\beta}_1\right)$ and $\text{Var}\left(\hat{\beta}_2\right)$ are, respectively, the variances of $\hat{\beta}_1$ and $\hat{\beta}_2$.

The heterogeneity of genetic effect size between high- and low-photodistributed site for per-SNP in the overall six studies is tested by fixed effect meta-analysis of

$$\hat{\delta}_i, \text{Var}\left(\hat{\delta}_i\right) \tag{9}$$

where $i$ is each of the six studies.

**Reporting summary.** Further information on research design is available in the Nature Research Reporting Summary linked to this article.

## Data availability

Data from 23andMe, Inc were made available under a data use agreement that protects participant privacy. Please contact dataset-request@23andme.com or visit research.23andMe.com/collaborate for more information and to apply to access the data. Precomputed rankings and P-values for the top 10,000 SNPs included in the GWAS meta-analysis are available in the figshare repository https://doi.org/10.6084/m9.figshare.11588325[51]. Any additional data (beyond those included in the main text and Supplementary Information) that support the findings of this study are available from the corresponding author upon request.

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

## Acknowledgements

We thank the research participants and employees of 23andMe for contributing to this work. In addition, we thank the participants and staff of the Nurses' Health Study and the Health Professionals Follow-up Study, for their valuable contributions, as well as the following state cancer registries for their help: A.L., A.Z., A.R., C.A., C.O., C.T., D.E., F.L., G.A., I.D., I.L., I.N., I.A., K.Y., L.A., M.E., M.D., M.A., M.I., N.E., N.H., N.J., N.Y., N.C., N.D., O.H., O.K., O.R., P.A., R.I., S.C., T.N., T.X., V.A., W.A. and W.Y. We assume full responsibility for analyses and interpretation of these data. EQTL data described in this paper were obtained from the GTEx Portal on 06/01/2018. We thank L. Tryggvadottir and G.H. Olafsdottir of the Icelandic Cancer Registry for assistance in the ascertainment of affected individuals. We thank D. Allain, V. Klee, and M. Bernhardt for ascertainment of cases and associated clinical data. The OSU Human Genetics Sample bank provided control samples. This work was supported in part by the National Human Genome Research Institute of the National Institutes of Health (grant number R44HG006981) and in part by NIH R01 CA49449, P01 CA87969, UM1 CA186107, UM1 CA167552, R03 CA219779, K23 CA211793 (K.Y.S.), and in part by Walther Cancer Foundation (J.H.). K.Y.S. is a Damon Runyon Clinical Investigator supported (in part) by the Damon Runyon Cancer Research Foundation. The Rotterdam Study is funded by Erasmus Medical Center and Erasmus University Rotterdam; Netherlands Organization for the Health Research and Development (ZonMw); the Research Institute for Diseases in the Elderly (RIDE); the Ministry of Education, Culture and Science, the Ministry for Health, Welfare and Sports, the European Commission (DG XII), and the Municipality of Rotterdam.

## Author contributions

K.Y.S. and J.H. designed the study. K.Y.S., Y.L., A.Z. G.T., R.D., A.R., L.P., S.N.S. and P.K. contributed data analyses. K.Y.S. and J.H. oversaw the study. K.Y.S., Y.L., R.D., A.Z. and S.N.S., contributed to the drafting of the paper. All authors, K.Y.S., Y.L., R.D., A.Z., G.T., A.R., L.M.P., W.W., P.A.K., A.U., T.N., A.E.T., J.H.O., B.S., K.T., E.J., A.S.W., P.K., S.N.S., K.S., M.M.A. and J.H critically reviewed the paper.

## Competing interests

G.T., S.N.S. and K.S. are employees of deCODE Genetics.
