## [Peer Review File · Nature Communications]

Reviewers' comments:

Reviewer #1 (Remarks to the Author):

The author performed a meta-analysis of Cu.SCC data. The number of sample that were analyzed was large, and reveal the identification of 8 novel foci associated with the disease. No validation or any functional studies were performed.

Reviewer #2 (Remarks to the Author):

This is a well written manuscript reporting novel findings from a GWAS of cutaneous squamous cell carcinoma (SCC). The research findings are interesting and well-presented, and the methods used in the study are appropriate. There are only a few minor points the authors may want to consider when revising the manuscript.

- 1) It is unclear whether principle components were adjusted for in the association analysis of SNPs with SCC risk.
- 2) It is mentioned several times in the manuscript text and tables that the newly-identified risk loci are involved in multiple protein-coding genes. It is unclear how these genes were identified. What analyses or criteria were used to identify these genes?
- 3) Lines 286-291: It appears that only three loci contain distinct association signals. Therefore, instead of stating that "the overall set of primary and secondary signals includes 20 common...", it would be more helpful to provide information on the secondary signals directly.
- 4) One risk variant has a MAF of <0.05 (line 292). Please provide additional information for this SNP, such as imputation quality by study.
- 5) Line 270: What does the phrase "SCC with effect 0.36" mean?
- 6) Is it possible to perform some analyses to evaluate potential interactions of sun exposure and certain risk variants or a polygenic score of a few relevant variants?

We appreciate the Reviewers' and Editor's thoughtful feedback on how to improve our manuscript. Please see our responses below. We believe we have addressed all issues and hope you will find our manuscript suitable for publication in *Nature Communications*.

Reviewer #1:

The author performed a meta-analysis of Cu.SCC data. The number of sample that were analyzed was large, and reveal the identification of 8 novel foci associated with the disease. No validation or any functional studies were performed.

Thank you for an accurate summary of our manuscript. We chose to include all datasets into a one-stage GWAS to maximize discovery potential rather than split into a two-stage GWAS. This international GWAS meta-analysis represents the largest GWAS for cutaneous SCC to-date. The findings reinforced all previously identified SNPs and identified 8 novel SNPs. Although there are no cellular-based functional studies included in this study, we analyzed RNA-sequencing data from paired-matched SCC and normal skin to identify loci associated with dysregulated gene expression and performed bioinformatics annotation of the SNPs. We also performed fine mapping to identify potential causal SNPs. Finally we performed a site-stratified subanalysis to identify SNPs associated with SCC in photo-distributed sites, suggesting a potential gene-environment interaction. Given the large scope of the study, we hope that you will find it acceptable for publication despite the lack of cellular functional studies.

Reviewer #2:

This is a well written manuscript reporting novel findings from a GWAS of cutaneous squamous cell carcinoma (SCC). The research findings are interesting and well-presented, and the methods used in the study are appropriate. There are only a few minor points the authors may want to consider when revising the manuscript.

Thank you for comment and the suggestions on how to improve our manuscript.

1) It is unclear whether principle components were adjusted for in the association

Thank you. All association studies were for adjusted principle components except the deCODE study. The deCODE data were adjusted differently because they use familial imputation for individuals who have not been directly genotyped, as outlined in reference 7. Variables adjusted for there include sex, county of birth, age, blood sample availability and the overlap of the lifetime of the individual with the timespan of phenotype collection. We have added an indicator of this information to the methods: page 4, paragraph 1:

“The methods used for association testing in each cohort have been described in detail.⁴⁻⁸. Briefly, association analysis was performed using logistic regression, assuming an additive model for allelic effects. Sex and population stratification were adjusted for by principal component (PC) analysis in each cohort except deCODE. The deCODE cohort was adjusted differently because it utilizes familial imputation for individuals who have

not been directly genotyped, as described.⁷ Five PCs were included to adjust for population stratification in the 23andMe and the NHS/HPFS cohorts. Ten PCs were adjusted for in the Kaiser and the Ohio cohorts. Rotterdam cohort adjusted for the four largest PCs. The linkage disequilibrium (LD) score regression was applied in the deCODE cohort to account for inflation in test statistics due to cryptic relatedness and stratification in the Icelandic population. the χ^2 statistics from GWAS scan were regressed against LD score and then the intercept was used as a correction factor.⁹”

2) It is mentioned several times in the manuscript text and tables that the newly-identified risk loci are involved in multiple protein-coding genes. It is unclear how these genes were identified. What analyses or criteria were used to identify these genes?

Association of genetic variant with closest protein coding gene was done by proximity.

We have clarified this in our methods on page 7 paragraph 1:

“We identified the closest related gene and evidence of regulatory function using HaploReg v4.1 (<http://archive.broadinstitute.org/mammals/haploreg/haploreg.php>).¹⁵ Gene annotations were based on the UCSC Genome Browser and GENCODE version 13. BEDTools was used to calculate the proximity of each variant to a gene by either annotation, as well as the orientation (3' or 5') relative to the nearest end of the gene, based on the strand of the gene.”

3) Lines 286-291: It appears that only three loci contain distinct association signals. Therefore, instead of stating that “the overall set of primary and secondary signals includes 20 common...”, it would be more helpful to provide information on the secondary signals directly.

Thank you for this suggestion. We have provided additional information on the distinct secondary signals in the three loci creating a new paragraph and streamlining the previous results. The new paragraph is on page 13 paragraph 2:

“Fine mapping revealed three loci with distinct secondary signals: rs6935510 at locus 6p25.3 ($r^2=0.12$ from the lead SNP rs12203592 in CEU population) is 2KB upstream of IRF4 in a predicted bivalent promoter region and alters a number of regulatory motifs; rs10962599, an intronic variant in BNC2 at 9p22.2, independent from lead SNP rs10810657 ($r^2=0.0012$ in CEU population) and in a H3K4me1 enhancer region in melanocytes, and rs4778138 at 15q13.1 which was independent from the lead SNP rs1800407 ($r^2=0.0032$ in CEU population). rs4778138 is in a novel locus and has been implicated in melanoma risk, hair and eye color.²⁴⁻²⁶”

4) One risk variant has a MAF of <0.05 (line 292). Please provide additional information for this SNP, such as imputation quality by study.

Thank you for that suggestion. rs117132860 has a MAF of 0.021 and was identified in prior GWAS studies and confirmed in our meta-analysis. We do not focus on this variant. We have added Supplementary Table 14 with imputation quality by study.

5) Line 270: What does the phrase “SCC with effect 0.36” mean?

Thank you for catching this. We have clarified this sentence:

“We examined the meta-analysis data and found that rs11571833 is associated with SCC with an effect size of 0.36 for the alternate (minor) allele and P value 1.0^{-6} , confirming the reported observation”

6) Is it possible to perform some analyses to evaluate potential interactions of sun exposure and certain risk variants or a polygenic score of a few relevant variants?

Unfortunately most of our cohorts do not have sun exposure information. We sought to address this by performing a subanalysis using cancer location site stratification data (stratifying into photo-distributed and non-photodistributed sites) to determine SNPs associated with SCC in photodistributed sites. We observed one SNP, rs721199, in which the T allele was specifically protective against SCC in low photodistributed sites (Supplementary Table 8). rs721199 is an eQTL in skin tissue for *HAL* which has been shown play a role in UV radiation mediated immunosuppression

While most of our cohorts did not have sun exposure data, sun exposure data was available for Harvard dataset (Nurses Health and Health Professional Cohorts). We have performed an interaction analysis of SNPs and Sun Exposure on SCC risk. No significant interaction was detected. The result is attached for your review but we have decided not to include in the manuscript as there were no interesting findings.

Methods:

To estimate sunlight exposure for each subject, a UV database for all 50 U.S. states was developed using reports from the Climatic Atlas of the United States, which reports mean daily solar radiation (in Langleys) at the earth's surface for weather stations around the country. Records of average annual solar radiation for January and July were extracted to represent winter and summer radiation, respectively. The mean solar radiation for each individual's past (at different age categories) and current residence was derived from the UV values measured at the nearest weather station, and both summer and winter radiation indices were developed for each residence. We also developed a cumulative lifetime sun exposure by combining the UV database and the information obtained from the supplementary questionnaires”

To evaluate interactions between sun exposure and genotypes, we modeled sun exposure level as a continuous variable using the median value among controls for each tertile in the Harvard cohort, which allowed us to assess the statistical significance of interaction by likelihood ratio tests.